# Bone Turnover Markers (CTX and P1NP) Following Low-Carbohydrate and Mediterranean Diet Interventions in Adolescents and Young Adults with Type 1 Diabetes

**DOI:** 10.3390/nu17243935

**Published:** 2025-12-16

**Authors:** Neriya Levran, Noah Levek, Yael Levy-Shraga, Noah Gruber, Rina Hemi, Ehud Barhod, Liana Tripto-Shkolnik, Arnon Afek, Efrat Monsonego-Ornan, Orit Pinhas-Hamiel

**Affiliations:** 1Pediatric Endocrine and Diabetes Institution, Edmond and Lily Safra Children’s Hospital, The Chaim Sheba Medical Center, Ramat Gan 52621, Israelyael.levy@sheba.health.gov.il (Y.L.-S.);; 2The Institute of Biochemistry, Food Science and Nutrition, The Faculty of Agriculture, Food and Environment, The Hebrew University of Jerusalem, Rehovot 9190501, Israel; 3Division of Nutrition Unit, The Chaim Sheba Medical Center, Ramat Gan 52621, Israel; 4National Juvenile Diabetes Center, Maccabi Health Care Services, Ra’anana 43450, Israel; 5Faculty of Medicine, Tel Aviv University, Tel Aviv 6997801, Israel; 6Endocrine Laboratory, The Chaim Sheba Medical Center, Ramat Gan 52621, Israel; 7Institute of Endocrinology, The Chaim Sheba Medical Center, Ramat Gan 52621, Israel; 8General Management, The Chaim Sheba Medical Center, Tel-Hashomer, Ramat Gan 52621, Israel

**Keywords:** low carbohydrate diet, mediterranean diet, type 1 diabetes, bone markers

## Abstract

**Background**: Impaired bone health is a recognized complication of type 1 diabetes. This study evaluated the effects of low-carbohydrate (LC) and Mediterranean (MED) diets on bone turnover markers in adolescents and young adults. **Methods**: In a 24-week randomized controlled trial, 40 individuals aged 12–21 years, with type 1 diabetes, were assigned to an LC or MED intervention (20 participants per group). *C*-terminal telopeptide (CTX) and procollagen type 1 *N*-terminal propeptide (P1NP) were measured at baseline and 24 weeks. **Results**: The groups had similar baselines. At 24 weeks, the between-group difference in delta glucose time in range was not statistically significant; median daily carbohydrate intake was 86 g (68–95) in LC and 130 g (102–173) in MED (*p* < 0.001). Comparing LC to MED, the median BMI z-score was lower (−0.1 [−0.3 to −0.1] vs. 0.0 [−0.1 to −0.1], *p* = 0.10), and calcium (*p* = 0.035) and magnesium intakes (*p* = 0.030) were lower. These associations did not remain statistically significant after false-discovery-rate correction. The median-adjusted alkaline phosphatase level decreased significantly in the LC group (*p* = 0.009). The median CTX changed following LC from 395 pg/mL (232–591) to 423 pg/mL (289–591) (*p* = 0.278); and following MED, from 357 pg/mL (244–782) to 296 pg/mL (227–661) (*p* = 0.245). P1NP changed in LC from 95 ng/mL (68–112) to 88 ng/mL (62–97) (*p* = 0.056) and in MED from 76 ng/mL (54–198) to 71 ng/mL (55–122) (*p* = 0.594). **Conclusions**: Exploratory analyses of bone turnover markers showed insignificant differences following LC and MED diets.

## 1. Introduction

The incidence of type 1 diabetes, one of the most common chronic autoimmune diseases in childhood, is rising worldwide [1,2]. Good glycemic control is the primary therapeutic goal in managing type 1 diabetes to prevent short- and long-term complications. Despite technological advancements and improved insulins, only a minority of the pediatric population with type 1 diabetes meets the glycemic goals of the International Society of Pediatric and Adolescent Diabetes [3]. Medical nutrition therapy is therefore a cornerstone of diabetes care [4].

Carbohydrate consumption has a pivotal effect on postprandial glycemic levels. Therefore, reducing carbohydrate intake lowers both glycemic response and insulin requirements. Recently, low-carbohydrate (LC) diets have gained popularity among individuals with type 1 diabetes as a strategy to manage blood glucose levels. However, adopting an LC diet may pose potential risks by excluding or limiting healthy carbohydrate sources and certain dairy products [5]. Moreover, the impact of LC diets on bone health should be carefully evaluated. Adequate dietary intake, particularly of calcium and vitamin D, is crucial to promoting optimal bone mass [6]. Evidence from a randomized controlled trial involving children and adolescents with type 1 diabetes suggests that supplementing diets with either milk or with pharmaceutical calcium combined with vitamin D may improve bone outcomes, particularly bone geometry [7].

Impaired bone health has become recognized as a chronic complication of type 1 diabetes [8]. Studies have consistently demonstrated lower bone-mineral density (BMD), disturbances in trabecular and cortical bone microarchitecture, and altered bone turnover in youth with T1D compared with healthy peers, together with an increased lifetime risk of fractures [9]. These skeletal deficits are multifactorial and arise from the combined effects of insulin and IGF-1 deficiency, chronic hyperglycemia, accumulation of advanced glycation end-products, low-grade inflammation, and disrupted muscle–bone crosstalk [9]. Linear growth and the accumulation of peak bone mass is particularly important in childhood and adolescence, whereas type 1 diabetes may negatively impact bone health [10]. Taken together, following an LC diet could potentially exacerbate harm to bones, by restricting certain foods that are beneficial for bone health.

*C*-terminal telopeptide (CTX) is a marker of bone resorption. Procollagen type 1 *N*-terminal propeptide (P1NP) is a biomarker of bone formation. These two bone turnover markers (BTMs) provide valuable insights into bone remodeling processes and can detect early changes in bone health that may not yet be visible on dual-energy X-ray absorptiometry (DEXA) scans [7].

We aimed to compare BTM, mineral balance, and overall nutritional statuses of adolescents and young adults with type 1 diabetes, following two six-month interventions: an LC diet or a Mediterranean (MED) diet. The LC diet restricted carbohydrate rich foods, whereas the MED diet emphasized plant-based foods, olive oil and nuts. This is part of a randomized trial comparing glycemic parameters following LC and MED diets. We focused particularly on the potential deficiencies or imbalances that could arise from restricting key food groups associated with bone health. To our knowledge, this is the first randomized controlled trial to examine the effects of dietary interventions, specifically an LC diet versus MED diet, on bone turnover markers in adolescents and young adults with type 1 diabetes.

## 2. Materials and Methods

### 2.1. Patients and Study Design

This was an open-label, parallel-group randomized controlled trial. The sample size was based on the primary endpoint, which was the delta of the time that the blood glucose was in the range of 3.9–10.0 mmol/L (70–180 mg/dL). We recruited 40 adolescents and young adults aged 12–21 years, who had been diagnosed with type 1 diabetes at least one year previously and who were using a continuous glucose monitoring system (Dexcom, Medtronic, or Libre). Of these, 70% were female. Exclusion criteria included a history of eating disorders or other mental illnesses in the participants or their immediate family, and a fracture in the past 12 months, as this could increase BTM and especially CTX [11]. Participants aged ≥18 years provided written informed consent. For participants aged <18 years, parents or legal guardians provided written informed consent, and the adolescents provided written assent. The study was approved by the Helsinki Committee at Sheba Medical Center protocol number (SMC-5537-18). Data of the participants’ medical history and medication use were obtained from their medical records

Participants were randomly assigned in a 1:1 ratio to either the LC diet group or the MED diet group. Randomization was conducted using computer-generated random permuted block sizes of 2 or 4, with group allocation concealed in sequentially numbered, sealed opaque envelopes that were opened only after enrollment. Socioeconomic position (SEP) was assessed using a national neighborhood-level index derived from data of the Israeli Central Bureau of Statistics, which combines information on education, income, employment, and material assets. SEP Index values are standardized continuous scores, and SEP Clusters represent deciles (1–10), with higher values indicating higher neighborhood socioeconomic position. Baseline clinical and demographic characteristics of the LC and MED groups were similar at enrollment (Table 1). Statistical analyses were conducted using IBM SPSS Statistics for Windows, Version 28 (IBM Corp., Armonk, New York, NY, USA, 2021).

### 2.2. Dietary Interventions

The LC diet consisted of 50–80 g of carbohydrates per day with no caloric restriction. The macronutrient composition was 15–20% carbohydrate, 33% protein, and 58% fat.

The MED diet consisted of a moderate-fat diet rich in vegetables, poultry, and fish, with limited red meat. The main sources of added fat were olive oil and nuts. The macronutrient composition was 40–50% carbohydrate, 25% protein, and 35% fat.

#### 2.2.1. Dietary Evaluation

Dietary intakes of dairy portions/day, calcium, phosphorus, zinc, and magnesium were assessed using a food frequency questionnaire (FFQ) at baseline and after 6 months. The FFQ comprised 116 food items commonly consumed in Israel, together with standard portion sizes. The questionnaire is adapted from a validated FFQ designed to assess the dietary intake of Israel’s multiethnic population [12].

#### 2.2.2. Dietary Assessment by Food Frequency Questionnaire

A trained dietitian completed the FFQ together with each participant in a face-to-face interview, covering all food items and clarifying frequencies and portion sizes. Nutrient intakes were calculated from the FFQ responses, and information on usual carbohydrate intake and meal patterns was corroborated using data downloaded from the participants’ insulin pumps (carbohydrate entries and bolus history).

### 2.3. Study Procedures

At baseline, each participant attended a group cooking workshop and an individual counseling session with a study dietitian, during which the assigned dietary plan was explained and personalized. Follow-up clinic visits with a dietitian were scheduled at weeks 1, 2, 4, 7, 10, 12, and 24. At each visit, dietary intake, glycemic data, and adverse events were reviewed, and individualized feedback was provided. Between visits, participants were asked to complete food diaries and 24 h dietary recalls, and to upload or share data from their diabetes management applications. These at-home tasks were explained verbally and summarized in printed handouts. Additional motivational phone calls were made twice during the first 12 weeks, to support adherence and address questions.

### 2.4. Bone-Turnover Markers

Blood was drawn from the cubital vein in the morning at baseline and after 12 h fasting, and after 6 months. BTM were measured using serum samples. All the assays were performed immediately after thawing using an automated analyzer, the iSYS (Immunodiagnostic Systems, plc, Tyne and Wear, UK), following the manufacturer’s instructions. The specific assays used were: CTX, measured with the IDS-iSYS CTX (CrossLaps) assay, and P1NP, measured with the IDS-iSYS intact P1NP assay. All the assays were chemiluminescence immunoassays, and a single batch was used for each assay to ensure consistency. Assay performance was validated using control specimens provided by the manufacturers. The intra- and inter-assay CV for CTX were 2.1–4.9% and 4.7–8.8%, respectively, and the intra- and inter-assay CVs for P1NP were 2.6–3 and 4.2–5.3%, respectively.

Percentage changes in CTX (pg/mL) and P1NP (ng/mL) were calculated between baseline and after the six-month intervention with the LC diet or MED diet, according to the formula:$$\mathrm{Percent}\;\mathrm{Change}\;=\;\left(\frac{6\;\mathrm{months}\;\mathrm{value}-\mathrm{baseline}\;\mathrm{value}}{\mathrm{baseline}\;\mathrm{value}}\right)\times 100$$

Changes reflecting 2.8 times the biological variation typically observed of the markers were considered significant. This was equivalent to a percentage change from baseline that exceeded the threshold of 30% for CTX and 21% for P1NP [13].

### 2.5. Anthropometric, Blood Pressure, and Pulse Measurements

At each visit, trained and certified staff followed a standardized protocol to measure height, weight, waist circumference, blood pressure, and pulse rate. Pubertal status was assessed by a pediatric endocrinologist using Tanner staging at baseline. Body weight and height were measured to the nearest 0.1 kg and 0.1 cm, respectively, using a balanced scale and Harpenden Stadiometer (Holtain Ltd., Crymych, UK). The body mass index (BMI) was determined by dividing the weight (in kilograms) by the square of the height (in square meters). BMI z-score norms were determined for individuals aged 12 to 20 years using CDC reference values. For participants who were over 20 years of age at the time of enrollment, we extrapolated BMI z-scores from the calculated BMIs at age 20 years [14]. BMI z-scores were classified as follows: normal body weight (−1.96 to 1, corresponding to percentiles 5.0–84.0), overweight (1–1.4, corresponding to percentiles 84.1–93.3), obesity (1.5–2, corresponding to percentiles 93.4–97.7), and morbid obesity (>2, above the 97.7 percentile) [15].

### 2.6. Biochemical Parameters

At each visit with the dietitian, blood samples were taken from fingertips and analyzed for ketones (specifically beta-hydroxybutyrate) using the Abbott Freestyle Optium Neo-Blood Glucose & Ketone Meter. Metabolic stability was ensured by assessing HbA1c, calcium, zinc, alkaline phosphatase, vitamin D, and magnesium. This was done from blood samples taken after at least 12 h of fasting, and provided that diabetic ketoacidosis was not present. The samples were collected at baseline and at 12- and 24-week intervals from a vein in the forearm and analyzed at the laboratories of Sheba Medical Center. Serum 25-hydroxyvitamin D was measured only at baseline.

### 2.7. Glycemic Parameters

HbA1c was measured at baseline and 6 months. Glycemic control was assessed from data accessed from continuous glucose monitors (CGMs) at each visit. The possible devices were the Dexcom CGM (Dexcom, San Diego, CA, USA), the Medtronic CGM (Medtronic, Northridge, CA, USA) and the Libre CGM (Libre, Alameda, CA, USA). Glycemic metrics included percentages of time: in range (3.9–10.0 mmol/L, 70–180 mg/dL), time above range (>10.0 mmol/L, >180 ng/dL), and time below range (<3.9 mmol/L, <70 mg/dL).

### 2.8. Statistical Analysis

The a priori sample size and power calculation was based on the primary endpoint, which was the delta of the time that the blood glucose was in the range of 3.9–10.0 mmol/L (70–180 mg/dL). To detect a difference of 20% with 80% power at a 5% significance level, with a two-sided paired t-test, 34 participants were needed to complete the study. To account for a 15% attrition rate, we included 40 participants. The sample size was calculated using PASS 2020 software.

Categorial variables were summarized as frequencies and percentages. Continuous variables were reported as medians and interquartile ranges (IQRs). For descriptive purposes, baseline continuous variables in Table 1 are additionally presented as arithmetic means with standard deviations (SDs). The Mann–Whitney test was used to compare variables between the two groups. The Chi square test and Fisher’s exact test were applied to compare categorical variables. Wilcoxon tests were used to compare continuous variables at the designated time points, and the McNamar test was used to compare categorical ones. Exploratory subgroup analyses were performed to examine changes in bone turnover markers stratified by sex using the Wilcoxon signed-rank test. For multiple within group comparisons, false discovery rate (FDR) adjusted *p* values were calculated using the Benjamini–Hochberg procedure.

According to an intention-to-treat approach, all the participants were analyzed as originally assigned. All the statistical tests were two-sided, and a *p*-value of <0.05 was considered statistically significant.

## 3. Results

### 3.1. Participants

Of the 40 participants with type 1 diabetes who participated, 28 (70%) were females. Twenty participants were randomized to the LC diet and 20 to the MED diet. Baseline clinical and demographic characteristics of the two groups are presented in Table 1. The cohort represented a population with established disease, with a median diabetes duration of 9.4 years (IQR 8.0–12.0) in the LC group and 10.0 years (IQR 7.0–14.5) in the MED group, ranging from 2 to 19 years overall. Thirty-eight participants were in Tanner stage 5, and 2 were in Tanner stage 3 (one in each treatment group). Based on BMI z-score categories, 55% of the participants in both the LC and MED groups were classified as having overweight or obesity.

### 3.2. Bone Turnover Markers

Bone formation and resorption were assessed using serum biomarkers. For the entire cohort, CTX levels were 7.5% lower than the normal range for age and sex. Similarly, P1NP levels were below the normal range for age and sex in 22.5% of participants.

Compared to baseline, after the six-month intervention, CTX levels increased slightly in the LC diet group and decreased in the MED diet group. These changes were not statistically significant (*p* = 0.208). P1NP levels showed a borderline significant decrease in the LC diet group (*p* = 0.056) and remained largely unchanged in the MED diet group (*p* = 0.594). The difference between the groups was not statistically significant (Table 2).

For the LC and MED groups, the median percentage changes from baseline to 6 months were +7.1% and −17.1%, respectively, for CTX; and −7.4% and −6.6%, respectively, for P1NP (Figure 1). These changes were all below the predefined thresholds for clinically meaningful percentage changes (Figure 1). The median alkaline phosphatase level decreased significantly over 6 months in both diet groups, yet the differences between the LC and the MED diet groups were not statistically significant (Table 2).

Subgroup analysis stratified by both diet and sex (Figure 2) indicated that the decline in P1NP among females was more pronounced in the LCD group (*p* = 0.052, *n* = 12) compared to the MED group (*p* = 0.34, *n* = 13).

### 3.3. Dietary Intake

At baseline, the median energy intake was 2537 kcal (1954–2773) in the LC diet group and 2077 kcal (1840–2661) in the MED diet group (*p* = 0.192) (Table 3). Compared to baseline, after six months, the energy intake was significantly lower in the LC diet group, 1533 kcal (1256–1758) (*p* < 0.001), while the decrease to 2050 kcal (1770–2827) in the MED diet group was not statistically significant. The median daily protein intake decreased significantly in the LC diet group, from 116 g to 91 g (*p* = 0.020), while remaining stable in the MED diet group 97 g to 98 g (*p* = 0.906). The median daily calcium intake decreased in the LC diet group, from 1315 mg to 966 mg (*p* = 0.063); and increased in the MED group, from 1283 mg/day to 1308 mg/day (*p* = 0.875), with no significant difference between the diets. The median daily magnesium intake decreased in the LC diet group, from 508 mg to 398 mg (*p* = 0.001), and increased in the MED diet group (*p* = 0.198), from 479 mg to 553 mg. The median phosphorus intake decreased in the LC diet group, from 1793 mg to 1462 mg (*p* = 0.010), while remaining stable in the MED diet group (*p* = 0.551). Serum 25-hydroxyvitamin D [25(OH)D] concentrations were at baseline were similar in the LC and MED groups (22 [IQR 18–30] vs. 24 [IQR 21–31] ng/mL, *p* = 0.916).

The portion of dairy servings was similar at baseline between the groups, with a median of 2.0 servings/day (1.0–3.0) in the LC group and 2.0 servings/day (0.0–2.35) in the MED group (*p* = 0.664). After six months, there was a significant difference in the change between the groups (*p* = 0.027), with the MED group increasing to a median of 3.0 servings/day (2.0–5.25), while the LC group remained at 2.0 servings/day (1.0–3.0). Within-group comparisons showed a significant increase in dairy servings in the MED group (*p* = 0.023), whereas no significant change was observed in the LC group (*p* = 0.443).

### 3.4. Anthropometric Outcomes

The median BMI z-score in the LC diet group decreased from 1.30 (0.65–1.58) to 1.26 (0.55–1.47) in six months. In the MED diet group, the median BMI z-score increased from 1.10 (0.61–1.42) to 1.24 (0.53–1.52) over the same period. The median BMI z-scores did not differ significantly between the groups at baseline (*p* = 0.529) or after six months (*p* = 0.968). The median changes from baseline were −0.06 (−0.33–0.07) and 0.03 (−0.08–0.14) in the respective groups (*p* = 0.07).

### 3.5. Glycemic Outcomes

The median time in range at baseline was 47% (34–55) for the LC diet and 52% (38–60) for the MED diet group (*p* = 0.211). After the 6-month intervention, the time in range increased to 58% (51–72) and 64% (47–71) in the respective groups (*p* = 0.989). The median delta values of time in range between baseline and 6 months were 16% (6–24) and 7% (−1–15) for the respective groups (*p* = 0.091). The median HbA1c levels at baseline were 8.1% (7.5–9.4) and 7.7% (6.6–8.2) respectively (*p* = 0.091). After six months, the median HbA1c levels were lower than at baseline: 7.2% (6.7–7.9) and 7.1% (6.6–7.8), respectively (*p* = 0.659) for the difference between the groups.

The median decrease in HbA1c from baseline was greater, for the LC than the MED diet group: −0.8% [−1.3–(−0.3)] vs. −0.2% (−0.8–0.175) (*p* = 0.026, FDR-adjusted *p* = 0.234). The reduction in HbA1c was significant in the LC diet group (*p* = 0.001), and not in the MED diet group (*p* = 0.453) (Table 2).

### 3.6. Correlations

Glycemic control- For the LC diet group, CTX and P1NP at 6 months were negatively correlated with delta time in range (r = −0.664, *p* = 0.003, adj. *p* value = 0.021) and (r = −0.741, *p* = 0.001, adj. *p* value = 0.014), respectively. HbA1c did not correlate with CTX (r = 0.128, *p* = 0.612, adj. *p* value = 0.778) or with P1NP (0.157, *p* = 0.548, adj. *p* value = 0.767). For the MED diet group, CTX and P1NP at 6 months were not correlated to delta time in range (r = 0.078, *p* = 0.744, adj. *p* value = 0.783) and (r = 0.074, *p* = 0.783, adj. *p* value = 0.783), respectively. CTX and P1Np in the MED group were not correlated with HbA1c levels.

Several other correlations (with carbohydrate intake, BMI z score, zinc and milk intake) appeared significant before adjustment, but none remained significant after FDR correction.

## 4. Discussion

In this open-label, randomized controlled trial of adolescents and young adults with type 1 diabetes, exploratory analyses of BTMs, including CTX and P1NP, did not show clear between-group differences over six months between the LC and MED diets. However, a negative trend toward lower P1NP levels was observed in the LC diet group, alongside reductions in the median alkaline phosphatase level following both dietary interventions. In our cohort, the proportions of participants with suboptimal bone markers, both before and after the intervention, were substantially higher than those reported in healthy adolescents aged 12–20 years in previous studies [16]. Differences in calcium and magnesium intake between groups favored the MED diet, underscoring its potential advantages for bone health.

### 4.1. LC Diet

During the intervention, 19 participants of the LC diet group significantly reduced their daily carbohydrate intake. This reduction led to a rebalancing of the remaining macronutrients, which in turn affected the intake of protein, as well as magnesium and calcium. These dietary changes may help explain the observed trend toward lower P1NP and higher CTX levels in the LC group, despite the absence of statistically significant between-group differences in BTMs. This pattern is consistent with previous reports that substantial carbohydrate restriction and weight loss may be associated with reduced bone formation and increased bone resorption markers in individuals following low-carbohydrate or calorie-restricted interventions [17].

### 4.2. Carbohydrates

The carbohydrate intake of the LC diet in our study was greater than that of most previous intervention studies that examined BTM following either a ketogenic or a modified ketogenic diet. Some studies showed no variation in BTM, while others reported significant changes [17,18]. For example, following a 3.5-week ketogenic low-carbohydrate, high-fat diet in elite endurance athletes, the fasted CTX concentration was increased by 22%, and the P1NP decreased by 14% after exercise [19]. Elsewhere, among 53 adults with epilepsy who followed a ketogenic diet for 12 weeks, decreases in CTX, P1NP and calcium levels occurred [20]. Although these studies involved different populations, had shorter durations, and focused on more stringent carbohydrate restrictions than our study, their findings may offer valuable insights into the potential effects of a long-term LC diet on BTM.

Among adults with type 1 diabetes, bone resorption was suppressed following glucose and insulin challenges with stronger effects observed when glucose was administered orally, likely due to incretin involvement [19]. However, Heikura et al., in this previous study focused primarily on short-term, immediate responses. In contrast, our study of the effects of an LC diet over six months offers insights into how sustained carbohydrate reduction and insulin adjustments impact bone metabolism in adolescents with type 1 diabetes. A possible physiological explanation for the association between carbohydrate intake and CTX levels is that the gut hormones might affect bone resorption markers. Jensen et al. found associations of postprandial responses of hormones such as insulinotropic polypeptide (GIP), glucagon-like peptide (GLP-1), and peptide YY (PYY), with CTX levels. Higher carbohydrate intake increases GIP and GLP-1, which are positively associated with CTX; and decreases PYY, which is inversely associated with CTX [21].

The specific impact on P1NP in individuals with type 1 diabetes is not as clearly established as for CTX. While some evidence suggests that hyperglycemia may impair bone formation, potentially leading to decreased P1NP levels, the relation remains complex. Factors such as insulin therapy, the duration of diabetes, and overall metabolic control can all play a role. Although carbohydrate intake likely affects P1NP levels, more research is needed to fully understand this relation in the context of type 1 diabetes [22].

The observed trends of lower P1NP and higher CTX may be attributable to reductions in both caloric and protein intake following the intervention, although the intake levels did not fall below the dietary recommended allowance. Protein intake is crucial for bone health, as it provides the essential amino acids necessary for collagen synthesis, which is directly reflected in P1NP levels, a marker of bone formation. Indeed, our findings demonstrate decreased protein intake in the LC group.

Protein supplementation has been shown to significantly enhance P1NP levels, particularly in populations with high bone turnover, such as adolescents and postmenopausal women [23,24]. Conversely, inadequate protein intake might lead to reduced P1NP levels, indicating impaired bone formation.

The effects on CTX, a marker of bone resorption, are more variable but tend to show a reduction with adequate protein supplementation, especially in older adults. This suggests that protein not only supports bone formation but may also help reduce bone loss [25]. In our study, the lower protein intake could have contributed to the trend toward higher CTX levels, reflecting increased bone resorption. The combined impact of reduced caloric and protein intake might therefore explain the trends observed in our participants, by which lower P1NP and higher CTX levels were noted, albeit the differences between the groups were not statistically significant. These findings underscore the importance of maintaining adequate protein intake to support bone health, particularly during interventions involving dietary modifications.

### 4.3. Body Weight

The difference between the intervention groups in the declines in BMI z-score was not statistically significant. However, four participants in the LC group experienced a weight decline of 7% or more; their P1NP levels were below the required threshold. Moreover, we report that lower weight following the LC diet was associated with higher CTX levels, thus suggesting an increase in bone resorption as weight decreases. Our finding corroborates Yu et al.’s finding that significant weight loss led to a decrease in P1NP and an increase in CTX in individuals, particularly females, with overweight and obesity following a calorie-restricted, high-protein, LC diet [26]. Our sex-stratified analysis indicated that the decline in P1NP among females was more pronounced in the LC diet than in the MED group. This observation warrants further investigation into the potential influence of distinct dietary patterns on bone metabolism in young women.

### 4.4. Minerals

In the LC diet group, zinc levels were significantly lower in the blood after the six-month intervention and were positively correlated to changes in CTX and P1NP. Zinc consumption was not significantly lower after the LC diet. Zinc is essential for the proper function of osteoblasts and synthesis of the bone matrix. Berger et al. showed that dietary zinc supplementation led to elevated P1NP levels in premenarcheal girls [27]. This suggests that the observed reduction in zinc intake in the LC diet group could potentially impact bone formation markers.

Magnesium intake decreased after the LC diet intervention. Although this did not affect blood levels, it may have affected BTM. Liu et al. demonstrated that magnesium supplements promote bone formation in rats and increased P1NP levels [28]. Magnesium on the other hand inhibits bone resorption. Among postmenopausal women, magnesium intake was found to be negatively correlated with CTX levels [29].

According to the 2022 Bone Health and Osteoporosis Foundation’s position statement, adequate calcium consumption is essential for achieving maximum bone density and preserving bone health throughout the life span [30]. We report significantly reduced calcium intake after six-months of the LC diet, which paralleled a decrease in milk intake. This contrasts with the MED diet group, which showed an increase in daily dairy servings. Dairy products are a major contributor to calcium intake. The positive correlation that was found between dairy servings and P1NP levels suggests a potential link between higher dairy consumption and bone formation. In a 12-week diet and exercise intervention in adolescent girls with overweight and obesity, bone resorption significantly decreased among those who consumed higher portions of dairy products (3–4 servings a day) [16].

### 4.5. Glycemic Control

We report that after six months of MED diet, HbA1c was negatively correlated with changes in CTX and positively correlated with the time that blood glucose was in the range of 3.9–10.0 mmol/L (70–180 mg/dL). This suggests that better glycemic control lowers bone resorption. Similarly, Madsen et al. reported a negative association of HbA1c with CTX, among 173 children and adolescents with type 1 diabetes [31].

The relation between metabolic control and BTM is influenced by various mechanisms. The demonstration that fasting hyperglycemia decreased CTX levels in prepubertal boys suggests that poor glycemic control affects the bone turnover response [32]. Elevated glucose levels also increase oxidative stress, which negatively impacts bone turnover [32]. El Amoutsy et al. found that children with T1D had lower P1NP levels and higher deoxypyridinoline levels, which correlated with higher oxidative stress markers, thus indicating impaired bone formation and increased bone resorption [33]. Lastly, an observational case–control study reported greater suppression of P1NP and CTX after an oral glucose tolerance test (OGTT) than after intravenous glucose infusion in nine people with T1D compared to healthy controls. This suggests a gut–bone axis. Additionally, individuals with T1D exhibit reduced suppression of CTX and inappropriate suppression of P1NP during OGTT, indicating a possible mechanism for increased bone fragility [34].

In the LC group, higher delta time in glucose range was significantly associated with lower CTX and P1NP levels at 6 months, indicating that improved glycemic stability was linked to reduced bone turnover. No such associations were found with HbA1c, and the MED group showed no correlations between glycemic changes and bone markers.

A critical factor in interpreting our findings is the disease duration within our cohort. Although the participants were adolescents and young adults, they represented a population with established disease. Chronic exposure to insulin deficiency and hyperglycemia over a decade drives the accumulation of advanced glycation end-products and structural skeletal deficits that may become ‘entrenched’ over time [10]. Consequently, bone metabolic set-points in individuals with long-standing type 1 diabetes may be less responsive to dietary modification than those in new-onset patients. While the six-month intervention period was sufficient to observe changes in acute metabolic parameters, it may have been too brief to reverse long-term pathophysiological adaptations in bone turnover markers.

Overall, significant reductions in several minerals were observed in the LC diet group. This could have implications for bone health, as these nutrients are crucial for maintaining bone integrity. In contrast, the MED diet group maintained more stable intake levels, with only minor decreases in mineral intakes. These findings underscore the distinct nutritional impacts of the two dietary interventions over the six-month period.

Taken together, our findings extend previous work on LC and ketogenic diets and bone health. In contrast to short-term ketogenic interventions in adults and athletes, which reported marked changes in CTX and P1NP [19]. Our 6-month LC and MED diets in adolescents and young adults with type 1 diabetes did not produce statistically significant between-group differences in bone turnover markers. However, the observed trends toward lower P1NP and higher CTX in the LC group, in the context of reduced protein, calcium, magnesium, and zinc intakes, are consistent with studies showing adverse effects of inadequate protein and mineral intake on bone turnover in youth and other high-risk populations [22,23,24,25,26,27,28]. Moreover, our correlations between glycemic control and CTX in the MED group align with previous reports linking poorer metabolic control with impaired bone turnover in children and adolescents with type 1 diabetes [31,32,33,34]. These parallels support the biological plausibility of our observations, while underscoring the need for larger, adequately powered trials to test these associations more definitively.

This study has several limitations that should be considered. The relatively short duration of the intervention, six months, may not have been enough to capture significant changes in BTM and long-term bone health. Additionally, we collected data on physical activity levels only at baseline, which substantially influences bone health. The use of an FFQ to assess dietary intake has inherent limitations, including potential recall bias and inaccuracies in estimating actual consumption. Furthermore, the small sample size limited the statistical power of the study and precluded analyzing gender-specific differences (preponderance of females), which are important for understanding bone health outcomes in adolescents with T1D. The age range (12–21 years) encompasses adolescents and young adults. BMI z-scores for participants aged >20 years were estimated by extrapolating age-20 reference values, which introduces some measurement uncertainties and may limit generalizability across age groups. In addition, the LC and MED diets were not isocaloric or iso-nitrogenous, and we were not able to fully adjust BTM analyses for concurrent changes in energy, protein, calcium, and magnesium intake. Moreover, vitamin D status and physical activity were assessed only at baseline. Therefore, some residual confounding by changes in vitamin D status and physical activity over time cannot be completely excluded. Finally, the large number of secondary and exploratory within- and between-group comparisons and correlation analyses increases the risk of type I error. Taken together, statistically significant findings from these analyses should be viewed as exploratory. Confirmation is required from larger, longer-term studies, specifically designed and powered to evaluate the skeletal effects of LC and MED diets in youth with type 1 diabetes.

## 5. Conclusions

Our findings highlight significant differences in nutrient intake between the diets. The observed trend toward lower P1NP levels in the LC diet group raises concerns about the potential impact of carbohydrate restriction on bone formation, which may be further influenced by an unexpected decline in protein intake observed in this group. Additionally, the correlations identified between nutrient intake, particularly calcium and zinc, and bone turnover markers underscore the critical role of a well-balanced diet in maintaining bone health in adolescents and young adults with type 1 diabetes. These findings are particularly relevant for clinicians prescribing dietary interventions for this population. We recommend monitoring dietary intake during LC diets and ensuring alignment with the nutrient targets and guidance provided by dietitians, with particular attention to calcium, protein, and other bone-supporting nutrients. Long-term studies that include baseline and follow-up bone density assessments are warranted to clarify the effects of defined dietary modifications on skeletal health in children, adolescents, and young adults with type 1 diabetes, especially considering their increased risk for fractures. Ultimately, personalized dietary strategies remain essential for managing type 1 diabetes-related complications and promoting overall health outcomes in adolescents and young adults with type 1 diabetes.

## Figures and Tables

**Figure 1 nutrients-17-03935-f001:**
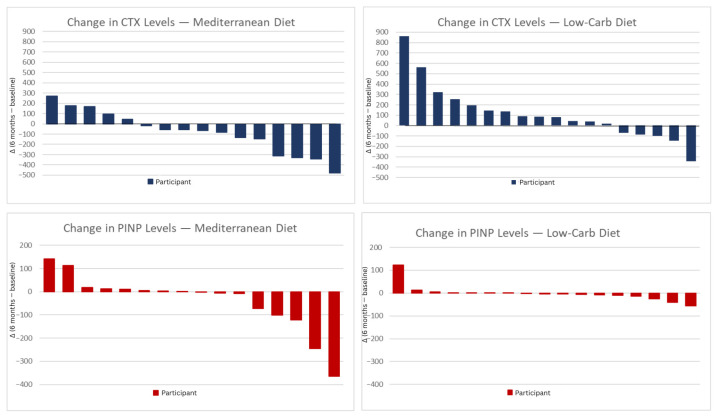
CTX and P1NP change after 6 months diet intervention. Individual changes in CTX and P1NP levels after 6 months of dietary intervention. The panels show changes between the 6-month and the baseline value, in CTX and P1NP for participants in the Mediterranean diet (**left**) and low-carbohydrate diet (**right**) groups. Each vertical bar represents one participant. Positive values indicate an increase and negative values a decrease in the respective biomarkers. MED; Mediterranean diet, LCD; low carbohydrate diet, CTX; *C*-terminal telopeptide, P1NP; procollagen type 1 *N*-terminal propeptide.

**Figure 2 nutrients-17-03935-f002:**
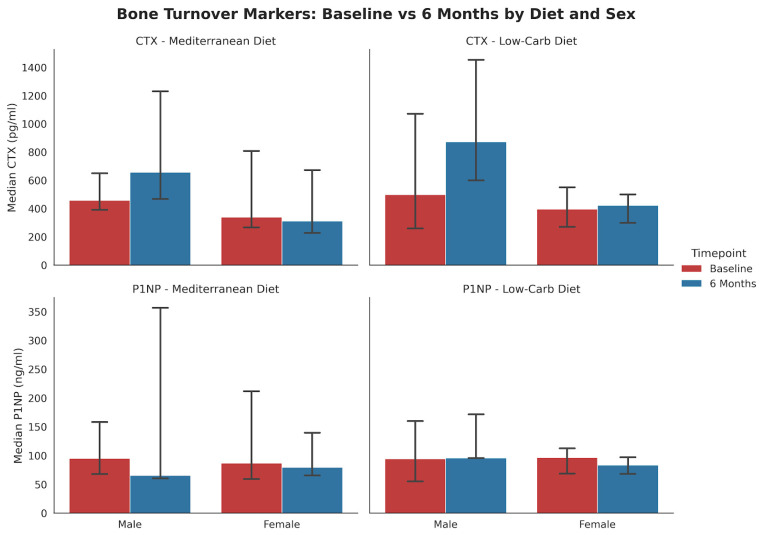
Changes in bone-turnover markers from baseline to 6 months stratified by diet and sex (median, IQR). The bar charts display mean serum concentrations of (**top**) CTX and (**bottom**) P1NP for the Mediterranean (MED) and Low-Carbohydrate (LC) diet groups. MED Diet: No statistically significant changes were observed in males (*n* = 3) or females (*n* = 13) for either marker. LC Diet: In female participants (*n* = 12), P1NP levels showed a borderline significant decrease (*p* = 0.052). Male participants (*n* = 5) showed no significant changes.

**Table 1 nutrients-17-03935-t001:** Baseline characteristics of the participants.

Characteristic	Low Carbohydrate Diet (*n* = 20)	Mediterranean Diet (*n* = 20)	*p*-Value
	Median (IQR), Mean ± SD	Median (IQR), Mean ± SD	
Age (year)	17.00 (15.00–19.00), 17.4 ± 2.8	18.00 (15.50–21.00), 18.1 ± 3.0	0.53
Female sex (%)	70	70	0.10
SEP Index	1.1 (0.9–1.7)	1.2 (0.9–1.7)	0.60
SEP Clusters	8.0 (7.2–9.0), 7.6 ± 1.7	8.0 (7.0–8.0), 6.7 ± 2.5	0.60
Diabetes duration (years)	10.0 (8.0–12.0), 9.4 ± 4.1	9.0 (7.0–14.5), 10.0 ± 4.7	0.93
BMI (kg/m^2^)	27.3 (22.7–28.3), 25.9 ± 4.5	25.5 (22.3–28.5), 25.6 ± 4.2	0.54
Waist circumference (cm)	85.7 (80.0–91.8), 85.1 ± 11.1	80.8 (72.2–90.7), 81.5 ± 11.7	0.62
Physical activity (hour/week)	1.50 (0.52–3.75), 1.9 ± 1.6	2.00 (0.00–4.00), 2.0 ± 1.8	0.90
Insulin pump treatment, *n* (%)	18 (90)	17 (85)	0.10
Time in range 70–180 mg/dL (%)	47 (34–55), 45.5 ± 12.3	52 (38–60), 49.8 ± 19.2	0.21
HbA1c (%)	8.1 (7.5–9.4), 8.2 ± 1.1	7.5 (6.9–8.6), 7.6 ± 1.0	0.09
mmol/mol	65 (58–79), 66 ± 12	58 (52–70), 59 ± 12	

Continuous variables are presented as median (interquartile range) and mean standard deviation. (Age is presented as median (interquartile range) and does not represent the full inclusion range of 12–22 years). All the categorical variable values are expressed as *n* (%). SEP Index, neighborhood socioeconomic position score; SEP Clusters, deciles of neighborhood socioeconomic position (1–10; higher values indicate higher SEP). Abbreviations: BMI, body mass index; HbA1c, hemoglobin A1c; SEP, socioeconomic position.

**Table 2 nutrients-17-03935-t002:** Biochemical data before and after the intervention of a low-carbohydrate diet (LCD) vs. Mediterranean (MED) diet.

	LCD	MED	*p* Between Groups	Adjusted *p*-Value
CTX (pg/mL)				
Baseline	395 (232–591)	357 (244–782)	0.984	0.989
6 months	423 (289–591)	296 (227–661)	0.208	0.596
Delta	62 (−83–141)	−69 (−318–114)	0.193	0.596
*p* value within groups	0.278	0.245		
Adjusted *p* value within groups	0.417	0.417		
P1NP (ng/mL)				
Baseline	95 (68–112)	76 (54–198)	0.854	0.953
6 months	88 (62–97)	71(55–122)	0.667	0.819
Delta	−4.4 (−12–1)	−0.2 (−106–11)	0.608	0.818
*p* value within groups	0.056	0.594		
Adjusted *p* value within groups	0.144	0.668		
HbA1c %				
mmol/L				
Baseline	8.1 (7.5–9.4)	7.7 (6.6–8.2)	0.091	0.596
	65.4 (58.4–79.0)	60.8 (48.6–66.0)		
6 months	7.2 (6.7–7.9)	7.1 (6.6–7.8)	0.659	0.818
	55.2 (49.5–63.3)	54.1 (48.6–61.9)		
Delta	−0.8 (−1.3–(−0.3))	−0.2 (−0.8–0.17)	0.026	0.234
	−32.7 (−37.2; −28.2)	−25.3 (−32.7–(−21.4))		
*p* value within groups	0.001	0.453		
Adjusted *p* value within groups	0.009	0.572		
Alkaline phosphatase IU/L				
Baseline	103.5 (85.7–151.0)	85.5 (74.5–169.2)	0.221	0.596
6 months	94.0 (80.5–112.5)	80.0 (68.2–141.2)	0.221	0.596
Delta	−11.5 (−33.2–(−3.0))	−5.50 (−27.2–(−0.5))	0.529	0.818
*p* value within groups	0.001	0.017		
Adjusted *p* value within groups	0.009	0.076		
Calcium 8.1–10.4 mg/dL				
Baseline	9.7 (9.6–10.1)	9.8 (0.6–10.0)	0.989	0.989
6 months	9.8 (9.5–9.9)	9.8 (9.5–10.1)	0.414	0.818
Delta	0.1 (−0.4–0.2)	0.0 (−0.2–0.2)	0.301	0.738
*p* value within groups	0.260	>0.999		
Adjusted *p* value within groups	0.417	1		
Zinc 50.0–150.0 mcg/dL				
Baseline	130.5 (104.7–150.0)	119.0 (96.5–140.0)	0.383	0.818
6 months	98.0 (83.5–119.0)	131.5 (110.5–150.7)	<0.001	0.001
Delta	−28.0 (−63.0–0.0)	8.5 (0.0–30.7)	<0.001	0.001
*p* value within groups	0.006	0.035		
Adjusted *p* value within groups	0.036	0.126		
Phosphorus 2.0–4.0 mg/dL				
Baseline	4.1 (3.8–4.4)	4.2 (3.4–4.3)	0.640	0.818
6 months	4.1 (3.7–4.4)	4.1 (3.5–4.3)	0.529	0.818
Delta	0.0 (–0.2–0.3)	−0.05 (−0.37–0.20)	0.529	0.818
*p* value within groups	>0.999	0.477		
Adjusted *p* value within groups	1	0.572		
Potassium 3.5–5.3 mmol/L				
Baseline	4.2 (4.2–4.3)	4.2 (4.1–4.4)	0.883	0.953
6 months	4.2 (4.1–4.4)	4.4 (4.2–4.6)	0.142	0.596
Delta	0.0 (0.0–0.2)	0.1 (−0.1–0.4)	0.565	0.818
*p* value within groups	0.124	0.044		
Adjusted *p* value within groups	0.279	0.132		
Magnesium 1.90–2.70 mg/dL				
Baseline	1.9 (1.8–2.0)	1.9 (1.8–2.0)	0.758	0.889
6 months	1.9 (1.8–2.0)	1.9 (1.8–2.0)	0.602	0.818
Delta	0.0 (−0.2–0.0)	0.0 (0.0–0.1)	0.175	0.596
*p* value within groups	0.261	0.424		
Adjusted *p* value within groups	0.417	0.572		

The data are presented as median (interquartile range). Delta indicates the change from baseline to 6 months. *p* Values within groups are derived from the Wilcoxon signed-rank test. False discovery rate (FDR)-adjusted *p* values for within group comparisons were calculated using the Benjamini–Hochberg procedure. Abbreviations: CTX, *C*-terminal telopeptide of type 1 collagen; P1NP, procollagen type 1 *N*-terminal propeptide; HbA1c, glycated hemoglobin A1c; IU, international units.

**Table 3 nutrients-17-03935-t003:** Changes in nutrient intakes over 6 months on a low carbohydrate diet versus a Mediterranean diet.

	LCD	MED	*p*-Value	Adjusted *p*-Value
Total energy kcal				
Baseline	2537 (1954–2773)	2077 (1840–2661)	0.192	0.508
6 months	1533 (1256;1758)	2050 (1770–2827)	<0.001	0.001
Energy percent ultra-process				
Baseline	16.6 (9.4–23.0)	17.7 (13.5–21.8)	0.529	0.662
6 months	11.1 (8.3–15.9)	15.2 (7.9–21.4)	0.289	0.529
Carbohydrate (gr)				
Baseline	265 (204–315)	173 (116;221)	0.211	0.508
6 months	86 (68–95)	130 (102;173)	<0.001	0.001
Protein (gr)				
Baseline	116 (92–138)	97 (82–121)	0.231	0.508
6 months	91 (69–108)	98 (79–126)	0.369	0.541
Fat (gr)				
Baseline	85 (78–108)	86 (72–105)	0.602	0.662
6 months	85 (65–96)	90 (70–105)	0.565	0.662
Fiber (gr)				
Baseline	33 (28–43)	27 (20–41)	0.174	0.508
6 months	21 (17–28)	29 (24–42)	0.009	0.066
Calcium (mg/day)				
Baseline	1315 (902–1468)	1283 (775–1640)	0.738	0.738
6 months	966 (635–1232)	1308 (899–1692)	0.035	0.154
Magnesium (mg/day)				
Baseline	508 (429–604)	479 (327–631)	0.640	0.670
6 months	398 (361–433)	553 (398–691)	0.030	0.154
Phosphorous (mg/day)				
Baseline	1793 (1615–2323)	1769 (1336–2038)	0.461	0.633
6 months	1462 (1181–1637)	1724 (1411–2024)	0.060	0.220
Potassium (mg/day)				
Baseline	4803 (3763–5768)	4304 (3338–5069)	0.341	0.535
6 months	3605 (2990–4316)	4110 (3248–5044)	0.231	0.535
Zinc (mg/day)				
Baseline	12.6 (11.0–16.0)	12.1 (9.9–14.1)	0.583	0.662
6 months	11.0 (8.3–12.4)	12.0 (9.8–15.6)	0.277	0.529

The data are presented as median (interquartile range). *p* Values between groups are derived from the Mann–Whitney U test. False-discovery-rate (FDR)-adjusted *p* values for between-group comparisons were calculated using the Benjamini–Hochberg procedure. “Energy percent ultra-process” indicates the percentage of total daily energy intake derived from ultra-processed foods (NOVA group 4). Abbreviations: LCD, low-carbohydrate diet; MED, Mediterranean diet.

## Data Availability

The data and the questioners are all in Hebrew and could be sent by a personal request.

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
