# Peer review of "Bone Turnover Markers (CTX and P1NP) Following Low-Carbohydrate and Mediterranean Diet Interventions in Adolescents and Young Adults with Type 1 Diabetes"

_nutrients, 2025, doi:10.3390/nu17243935_

Round 1
Reviewer 1 Report
Comments and Suggestions for Authors
The trial was powered for a glycemic endpoint (time-in-range), not for bone turnover markers (BTMs). With n=20 per arm over 6 months, the null findings on CTX/P1NP are uninterpretable as evidence of “no effect.” Drawing bone-health inferences from secondary, underpowered endpoints is not supportable.
Between-group differences in time-in-range did not reach significance, yet the narrative leans on HbA1c and exploratory correlations and then segues to bone markers. This reads like outcome switching/“spin” rather than a confirmatory test of the a priori endpoint.
Despite “no caloric restriction,” the LC arm cut energy intake by ~1,000 kcal/day and significantly lowered protein, calcium and magnesium, each independently affects BTMs. Any LC-vs-MED inference on bone metabolism is confounded by these non-isocaloric, non-iso-nitrogenous changes.
Adolescents’ BTMs are highly sensitive to Tanner stage, 25(OH)D status, and activity. None were measured/adjusted. Without them, BTMs in a 12–22 y mixed cohort are too noisy to interpret. (Physical activity is acknowledged as missing; vitamin D and pubertal staging are not handled.)
Six months of BTMs without DXA/HR-pQCT, bone geometry, or fracture risk cannot support clinical claims or practice recommendations (e.g., “assess bone density before LC”). That conclusion overshoots the evidence.
Dozens of within- and between-group tests plus subgroup correlations (e.g., dairy ↔ P1NP, zinc ↔ BTMs) are presented with p≈0.03–0.05 and no clear multiplicity control strategy—high risk of false positives.
The age span (12–22) mixes adolescents with young adults; BMI z-scores for >20 y were “extrapolated,” adding measurement uncertainty. The sample is 70% female; no sex-stratified analysis is feasible with n=20/arm.
Narrative links dietary patterns → gut hormones/micronutrients → bone remodeling, but the study provides no direct mechanistic measures (incretins, bone formation/resorption dynamics beyond two BTMs), and makes policy-like recommendations.
Author Response
Author response letter
We appreciate the comments that we received, which have helped us to prepare a clearer and more comprehensive documentation and interpretation of our findings.
Reviewer #1
Item# 1: The trial was powered for a glycemic endpoint (time-in-range), not for bone turnover markers (BTMs). With n=20 per arm over 6 months, the null findings on CTX/P1NP are uninterpretable as evidence of “no effect.” Drawing bone-health inferences from secondary, underpowered endpoints is not supportable.
Response: We thank the reviewer for this important remark. In line with this concern, we have revised the Abstract, Results, Discussion and Conclusions to avoid any language that could be interpreted as evidence of “no effect” on bone metabolism. We now explicitly state that the BTM analyses are exploratory and underpowered, and that the null findings cannot exclude potential dietary effects on bone. The conclusions have been appropriately tempered to emphasize that the BTM findings are hypothesis-generating and that larger, longer-term trials specifically powered for skeletal outcomes are required. We have also added a statement to the Limitation section noting that the study was underpowered for bone endpoints. Nevertheless, we believe that the findings remain relevant for families, caregivers, and clinicians caring for individuals following LC diets, and that further studies are needed to confirm and expand upon these results.
Specific revision made in the manuscript include:
Abstract, Results - We changed the text as follows: "After LC compared to MED, the median BMI z-score was lower (−0.1 [−0.3 to −0.1] vs 0.0 [−0.1 to −0.1], p=0.10), and calcium (p=0.035) and magnesium intakes (p=0.030) were lower. These associations did not remain statistically significant after false discovery rate correction. The median adjusted alkaline phosphatase level decreased significantly in the LC group (p=0.009). (Page 1, lines 32-37)
Conclusions: We changed the text as follows: “Exploratory analyses of bone turnover markers showed insignificant differences following LC and MED diets." (Page 1, lines 41-420
Discussion-
Revised: “In this open-label randomized controlled trial of adolescents and young adults with type 1 diabetes, exploratory analyses of BTMs, including CTX and P1NP, did not show clear between-group differences over six months, between the LC and MED diets" (Page 11 lines 362-364).
Limitations- we added the following text-
"Taken together, statistically significant findings from these analyses should be viewed as exploratory. Confirmation is required from larger, longer-term studies, specifically designed and powered to evaluate the skeletal effects of LC and MED diets in youth with type 1 diabetes.” (Page 14 line 521-525)
Item#2: Between-group differences in time-in-range did not reach significance, yet the narrative leans on HbA1c and exploratory correlations and then segues to bone markers. This reads like outcome switching/ “spin” rather than a confirmatory test of the a priori endpoint.
Response: We appreciate the reviewer’s comment and agree that the hierarchy between primary and secondary/exploratory outcomes must be clearly conveyed to avoid any impression of “spin”.
We thank the reviewer for noting that our analysis using HbA1c was not aligned with the study’s primary glycemic outcome. In response, we performed correlation analyses using delta time-in-range instead. These analyses revealed significant associations with bone turnover markers in the LC group. Accordingly, we revised the Discussion to reflect these findings: "In the LC group, higher delta time in glucose range was significantly associated with lower CTX and P1NP levels at 6 months, indicating that improved glycemic stability was linked to reduced bone turnover. No such associations were found with HbA1c, and the MED group showed no correlations between glycemic changes and bone markers." (Page 13 lines 482-484 )
As requested, we changed in the result section the correlations with the primary outcome, delta time in range (Page 8, line 322-324).
We added to the Conclusions the following: "We recommend monitoring dietary intake during LC diets and ensuring alignment with the nutrient targets and guidance provided by dietitians, with particular attention to calcium, protein, and other bone-supporting nutrients. Long-term studies that include baseline and follow-up bone density assessments are warranted to clarify the effects of defined dietary modifications on skeletal health in children and adolescents with type 1 diabetes, especially considering their increased risk for fractures." (Page 14 lines 534-540)
We believe these changes render the analytical hierarchy explicit and ensure presentation of the confirmatory test of the priori endpoint.
Item# 3: Despite “no caloric restriction,” the LC arm cut energy intake by ~1,000 kcal/day and significantly lowered protein, calcium and magnesium, each independently affects BTMs. Any LC-vs-MED inference on bone metabolism is confounded by these non-isocaloric, non-iso-nitrogenous changes.
Response: We thank the reviewer for this important clarification. We agree in the LC group, the total energy intake decreased substantially during the intervention, despite the diet not being prescribed as calorie-restricted. This change was accompanied by lower intakes of protein, calcium, and magnesium, all of which are known to independently influence bone turnover markers. Therefore, as the reviewer correctly notes, any LC–versus–MED comparison of CTX and P1NP is inherently confounded by these non-isocaloric and non-iso-nitrogenous differences.
To address this, we have revised the Results, Discussion, and Conclusions to explicitly state that the observed trends in BTMs cannot be attributed solely to carbohydrate restriction, and may instead reflect the combined effects of lower energy and micronutrient intake. As stated above, we now clearly frame the BTM findings as exploratory and hypothesis-generating rather than as causal inferences regarding LC diets.
In line with your suggestion regarding the potential implication of causality, we revised the manuscript title to remove the term "impact". The title now reads: "Bone turnover markers (CTX AND P1NP) following low carbohydrate diet and Mediterranean diet interventions in adolescents with type 1 diabetes" (Page 1 lines 2-3).
To clarify the matter in the manuscript, we have strengthened the Limitations section as follows: "In addition, the LC and MED diets were not isocaloric or iso-nitrogenous, and we were not able to fully adjust BTM analyses for concurrent changes in energy, protein, calcium and magnesium intake(Page 14 line 515-517)
Item#4 : Adolescents’ BTMs are highly sensitive to Tanner stage, 25(OH)D status, and activity. None were measured/adjusted. Without them, BTMs in a 12–22 y mixed cohort are too noisy to interpret. (Physical activity is acknowledged as missing; vitamin D and pubertal staging are not handled.)
Response: We thank the reviewer for raising this important point. At baseline, we collected data on pubertal status, vitamin D, and physical activity. Pubertal status was assessed using Tanner staging. The vast majority of the participants were in Tanner stage 5, and only two participants were in Tanner stage 3 (one in the LC group and one in the MED group; p = 1.00). This indicates very limited variability and no imbalance between groups. We have now added this information on Tanner stage to the Methods and Results sections.
Serum 25-hydroxyvitamin D [25(OH)D] concentrations were also measured at baseline and were similar in the LC and MED groups (22 [IQR 18–30] vs. 24 [21–31] ng/mL, p = 0.916). These data are now reported in the in the methods: ". Serum 25-hydroxyvitamin D was measured only at baseline. " (Page 4 line 192) and in the results: Serum 25-hydroxyvitamin D [25(OH)D] concentrations were at baseline were similar in the LC and MED groups (22 [IQR 18–30] vs. 24 [21–31] ng/mL, p = 0.916) (Page 7 lines 291-293). Because vitamin D status is only minimally influenced by short-term dietary changes and baseline values did not differ between the groups, we did not repeat 25(OH)D measurements at the end of the follow-up.
Physical activity was assessed at baseline by self-report, and baseline activity levels did not differ significantly between the groups. Although physical activity was not reassessed during follow-up, there is no indication that activity patterns changed differentially between the groups over the 6-month period. We have added the baseline physical activity in Table 1.
We have revised the Methods and Results to specify that Tanner stage, 25(OH)D, and physical activity were assessed at baseline and were comparable between the groups. In the Limitations section, we also note that these variables were measured only once, that pubertal stage showed very limited variability in our cohort, and that physical activity was assessed using a relatively crude self-report measure. Therefore, some residual confounding by growth, vitamin D status, and physical activity over time cannot be completely excluded.
Methods (Page 4 lines 172-173): “Pubertal status was assessed by a pediatric endocrinologist using Tanner staging at baseline.
Results (Page 5 lines 225-226) "…38 were in Tanner stage 5 and 2 were in Tanner stage 3 (one in each treatment group)."
Physical activity was added to table 1. (Page 6 lines 240)
Limitation (Page 14 line 521-524)
"Moreover, vitamin D status and physical activity were assessed only at baseline. Therefore, some residual confounding by changes in vitamin D status and physical activity over time cannot be completely excluded."
Item#5: Six months of BTMs without DXA/HR-pQCT, bone geometry, or fracture risk cannot support clinical claims or practice recommendations (e.g., “assess bone density before LC”). That conclusion overshoots the evidence.
Response: We agree that six months of BTM data, without DXA/HR-pQCT, structural bone measures, or fracture outcomes, cannot support guideline-level practice recommendations. Our intention was not to propose a formal recommendation, but to draw attention to two signals observed in this cohort: (1) a relatively high prevalence of suboptimal bone markers in adolescents and young adults with type 1 diabetes at baseline, and (2) Reductions in calcium, magnesium, and protein intake among those adopting an LC diet.
In line with the reviewer’s concern, we have revised the Discussion and Conclusions to remove phrasing that could be interpreted as a directive to “assess bone density before LC”. Instead, we framed our message more cautiously, emphasizing vigilance and individual risk assessment rather than a universal practice recommendation. Specifically, we now state: "We recommend monitoring dietary intake during LC diets and ensuring alignment with the nutrient targets and guidance provided by dietitians, with particular attention to calcium, protein and other bone-supporting nutrients. Long-term studies that include baseline and follow-up bone density assessments are warranted to clarify the effects of defined dietary modifications on skeletal health in children and adolescents with type 1 diabetes, especially considering their increased risk for fractures." (Page14, line 541-544)
Item#6 Dozens of within- and between-group tests plus subgroup correlations (e.g., dairy ↔ P1NP, zinc ↔ BTMs) are presented with p≈0.03–0.05 and no clear multiplicity control strategy—high risk of false positives.
Response: We thank the reviewer for this important comment. To reduce the risk of false positives arising from the large number of within- and between-group tests and subgroup correlations, we applied the Benjamini–Hochberg false discovery rate (FDR) procedure (q = 0.05) within each exploratory analytic family (nutrient–BTM correlations and blood biochemistry variables). Following FDR adjustment, several associations with p-values in the 0.03–0.05 range (e.g., dairy intake, zinc, BMI z-score) no longer met the significance threshold and are now reported and interpreted as exploratory. The Methods and Results sections, as well as all the relevant tables, have been revised to reflect the FDR-adjusted p-values and the exploratory nature of these findings. In addition, we revised the Statistical Analysis section to describe the FDR procedure, updated the relevant tables to include adjusted p-values and explanatory footnotes, and modified the Results section to clearly present the FDR-corrected findings.
Methods: " For multiple within group comparisons, false discovery rate (FDR) adjusted p values were calculated using the Benjamini-Hochberg procedure." (Page 5 line 216-217).
Results: Table 2 and 3 (Pages 8-11)
Item#7 The age span (12–22) mixes adolescents with young adults; BMI z-scores for >20 y were “extrapolated,” adding measurement uncertainty. The sample is 70% female; no sex-stratified analysis is feasible with n=20/arm.
Response: Indeed, the age range of 12–21 years encompasses both adolescents and young adults, and the sample is predominantly female. This reflects the real-world population of our pediatric and transition clinic, in which many patients are in late adolescence or early adulthood. As noted in the revised Methods and Results, most of the participants were fully pubertal (Tanner stage 5), and the median age in both groups was in the older adolescent range. The latter reduces, though does not eliminate, developmental heterogeneity.
For participants aged ≤20 years, BMI z-scores were calculated using age- and sex-specific reference data. For those aged >20 years, BMI z-scores were extrapolated from the reference values at age 20. We have clarified this procedure in the Methods and now emphasize that BMI z-scores were used primarily to describe weight status at the group level rather than as a primary outcome. We also acknowledge in the Limitations that extrapolation for participants aged >20 years introduces some measurement uncertainty.
We agree that the small sample size precludes adequately powered sex-stratified analyses. We have added this point explicitly to the Limitations, noting the preponderance of females, and that our findings may not fully generalize to males. Future studies with larger and more balanced samples are needed to explore potential sex differences in diet–bone relations. (Page 14 lines 515)
We added this sentence to the limitation section: "The age range (12–21 years) encompasses adolescents and young adults. BMI z-scores for participants aged >20 years were estimated by extrapolating age-20 reference values. This introduces some measurement uncertainty and may limit generalizability across age groups." (Page 14 lines 516-519)
Item#8 Narrative links dietary patterns → gut hormones/micronutrients → bone remodeling, but the study provides no direct mechanistic measures (incretins, bone formation/resorption dynamics beyond two BTMs), and makes policy-like recommendations.
Response We appreciate this point. In this trial we did not measure gut hormones, detailed bone microarchitecture, or dynamic bone formation/resorption beyond CTX and P1NP. We agree that our data cannot provide direct mechanistic proof for a dietary pattern → incretins/micronutrients → bone remodeling pathway. Our intention in that section was not to claim that these mechanisms were demonstrated by our study. Rather, we addressed the observed patterns in BTMs and nutrient intake within the context of existing literature on incretins, glucose excursions, and bone turnover, and offered a plausible physiological framework.
To avoid any over-interpretation, we have revised the Discussion to clearly describe the mechanistic links (gut hormones, micronutrients, bone remodeling) as hypothesized pathways based on prior evidence, and not as mechanisms directly tested in our data. Moreover, we explicitly stated that CTX and P1NP were the only bone turnover markers measured, and that no direct hormonal or structural bone measures were obtained in this trial.
Regarding “policy-like recommendations”, we agree that the original phrasing about “assessing bone density before LC” could be understood to indicate stronger guidance than our data support. We have therefore replaced that sentence with more cautious wording in the Conclusions, focusing on monitoring and future research rather than on a prescriptive recommendation. Specifically, we now state the following: “We recommend monitoring dietary intake during LC diets and ensuring alignment with the nutrient targets and guidance provided by dietitians, with particular attention to calcium, protein, and other bone-supporting nutrients. Long-term studies that include baseline and follow-up bone density assessments are warranted to clarify the effects of defined dietary modifications on skeletal health in children and adolescents with type 1 diabetes, especially considering their increased risk for fractures.” (Page 14 lines 538-544)
Taken together, the revised text presents our mechanistic discussion as contextual and hypothesis-generating, and our clinical message as one of vigilance and the need for further research, rather than as a basis for formal practice recommendations.
Reviewer 2 Report
Comments and Suggestions for Authors
Interesting idea of this study, my recommendations are the following:
Abstract – I recommend mentioning the number of subjects in the two study groups. Mention the results of the BMI z-scores, but according to the study, only subjects over 20 years old had this indicator calculated, consequently I recommend clarifications, especially since the subjects are between 12-22 years old. Also the z-score is within normal limits, in both groups and in both tests.
Key findings I recommend mentioning the typology of the subjects-adolescents.
Introduction – I recommend expanding this section by mentioning the clinical and metabolic aspects of the characteristics of the target group, focusing on information for adolescents. I also recommend mentioning the research gaps regarding the purpose of the study. I recommend expanding the aspects regarding the nutritional characteristics of the two diets selected in the study in correlation with the specifics of the subjects.
Methods – section 2.1. - I recommend mentioning whether informed consent was obtained from subjects over 18 years of age to participate in the study. I recommend mentioning the ethics committee number and the date on which the registration was made. I recommend mentioning the typology of the study. I recommend clarifying the aspects regarding the homework received in the envelopes. I recommend calculating the sample power.
I recommend introducing a new section called Procedures where the aspects regarding the homework sent in the envelopes, the meetings with the dieticians and the relevant aspects of these meetings should be mentioned.
Section 2.2. I recommend reorganizing it into two subsections: intervention and evaluation questionnaire. FFQ – I recommend mentioning the method of completing the questionnaire, physically, online and the method of making the answers.
2.4. I recommend mentioning the instruments used in the evaluation of anthropometric measurements.
Lines 133-135 it is not clear what you want to mention. The z-score was calculated only for people over 20 years old, what is their number in each group, the other subjects were not registered with this score although it is also mentioned in the abstract under Results, I recommend clarification.
2.6. I recommend mentioning the testing instrument used.
Lines 156-161 I recommend moving to section 2.1.
Lines 178-177 I recommend moving to section 2.1.
I recommend mentioning in the Statistical Analysis section, which should be numbered as part of the Methods section, also the statistical indicators mentioned in table 1, i.e. the arithmetic mean. I also recommend calculating SD – standard deviation.
Table 1 mentions the age ranges for the LC group, the age is between 15-19 years and for the MED between 15.5-21, it is not clear why until you mention that the age was between 12-22 years, I recommend clarification. Also, if the arithmetic mean was calculated for the parameters in question, I recommend mentioning two decimal places as well as the standard deviation. SEP Index; SEP Clusters – these two parameters are not mentioned in the previous text in the Methods section, I recommend clarification. I recommend that all acronyms be mentioned descriptively under table 1.
I think it is a major mistake to mention twice in table 1 - Diabetes duration (years), with different values, I recommend clarification.
SEP, socioeconomic 207 position – I recommend that this aspect be detailed and clarified.
Fig 1 , I recommend mentioning what the vertical and horizontal axes represent.
Table 2 recommends that the information presented in this table should not be duplicated when interpolating, I recommend revision.
Lines 365-368 I think it is an error, you refer strictly to the results of this study, but still mention the bibliographic index 15, which has a different typology, I recommend clarifications.
4.1. I recommend clarifications, so far you have presented the results per group, and in this very short section, per individuals. Also, this subsection being part of the Discussion section, I recommend that correlations be made between the results recorded with results from previous studies.
I recommend revising the Discussion section by making concrete correlations between the results of this study with results from previous studies, focused.
I recommend that the conclusions be revisions, focused on the results.
Lines 492-494 I recommend moving to the Introduction section, it is not a conclusion.
Author Response
Reviewer 2: Interesting idea of this study, my recommendations are the following:
We thank the reviewer for the thoughtful and constructive comments. Below we address each point in turn and indicate the corresponding changes made in the manuscript.
Item#1 Abstract – I recommend mentioning the number of subjects in the two study groups.
Response: We agree and now explicitly report the number of participants per group in the Abstract. “…40 individuals aged 12–21 years, with type 1 diabetes, were assigned to an LC or MED intervention (20 participants per group).” (Page 1 line 28)
Item#2 mention the results of the BMI z-scores, but according to the study, only subjects over 20 years old had this indicator calculated, consequently I recommend clarifications, especially since the subjects are between 12-22 years old. Also, the z-score is within normal limits, in both groups and in both tests.
Response: We thank the reviewer for this remark. As detailed in the Methods, BMI z-scores were calculated for all the participants aged 12–21 years. For participants aged ≤20 years, age- and sex-specific norms were used. For those aged >20 years, BMI z-scores were extrapolated from the reference values at age 20 years. This is consistent with previous publications [Lumeng, J.C.; Kaciroti, N.; Frisvold, D.E. Changes in Body Mass Index z Score over the Course of the Academic Year among Children Attending Head Start. Acad Pediatr 10, 179–186, doi:10.1016/j.acap.2010.01.008.]). Importantly, in our cohort, BMI z-scores did not classify all the participants within the normal range, and a substantial proportion in both groups were classified with overweight or obesity. To clarify this point, we have revised the text slightly in the Methods and Results, without adding extensive anthropometric detail, as the primary focus of this manuscript was bone turnover markers.
Changes in the manuscript:
Methods, Section 2.4: “…For participants who were over age 20 years at the time of enrollment, we extrapolated BMI z-scores from the calculated BMIs at age 20 years [15]. BMI z-scores were classified as follows: normal body weight (−1.96 to 1…), overweight (1–1.4…), obesity (1.5–2…), and morbid obesity (>2…)].” (Page 4 line 178-181)
Results, baseline characteristics: “Based on BMI z-score categories, 55% of the participants in both the LC and MED groups were classified as having overweight or obesity. (Page 5 lines 27-228)
Item#3 Key findings I recommend mentioning the typology of the subjects-adolescents.
Response: We agree and have clarified the population in our summary statements by explicitly referring to adolescents and young adults with type 1 diabetes. ( Page 1 line 26)
Item#4 Introduction – I recommend expanding this section by mentioning the clinical and metabolic aspects of the characteristics of the target group, focusing on information for adolescents. I also recommend mentioning the research gaps regarding the purpose of the study. I recommend expanding the aspects regarding the nutritional characteristics of the two diets selected in the study in correlation with the specifics of the subjects.
Response: We appreciate this suggestion and have expanded the Introduction to better describe the clinical and metabolic characteristics of adolescents with type 1 diabetes, including their vulnerability to impaired bone accrual. We also now stated more explicitly the research gaps regarding the skeletal effects of LC versus MED diets in this population, and summarized the key nutritional features of these two dietary patterns that are relevant to bone health in youth.
We added the following sentences to the introduction including new reverences:
“Studies have consistently demonstrated lower bone mineral density (BMD), disturbances in trabecular and cortical bone microarchitecture, and altered bone turnover in youth with T1D compared with their healthy peers, together with an increased lifetime risk of fractures (9). These skeletal deficits are multifactorial and arise from the combined effects of insulin and IGF-1 deficiency, chronic hyperglycemia, and the accumulation of advanced glycation end-products, low-grade inflammation, and disrupted muscle–bone crosstalk. (Page 2 line 65-70)
And key nutritional features of these two dietary patterns:
The LC diet restricted carbohydrate rich foods, whereas the MED diet emphasized plant-based foods, olive oil and nuts. (Pages 2 lines 82-83)
Item#5 Methods – section 2.1.
Response: We thank the reviewer for these detailed and helpful suggestions regarding the Methods and baseline anthropometric description. We have revised the manuscript accordingly, as detailed below.
Item#6 I recommend mentioning whether informed consent was obtained from subjects over 18 years of age to participate in the study. I recommend mentioning the ethics committee number and the date on which the registration was made.
Response: In the revised version, we clarified the consent process for adults and minors and provide the ethics committee number and trial details. The updated text reads:
“Participants aged ≥18 years provided written informed consent. For participants aged <18 years, parents or legal guardians provided written informed consent and the adolescents provided written assent. The study was approved by the Helsinki Committee at Sheba Medical Center (protocol number SMC-5537-18) " (Page 3, lines 101-104).
Item#7 I recommend mentioning the typology of the study.
Response: we added: "This was an open-label, parallel-group randomized controlled trial.” (Page 3 line 93).
Item#8 I recommend clarifying the aspects regarding the homework received in the envelopes.
Response: To avoid any confusion with “homework,” we clarified that the envelopes were used solely for allocation concealment:
“Participants were randomly assigned in a 1:1 ratio to either the LC diet group or the MED diet group. Randomization was conducted using computer-generated random permuted block sizes of 2 or 4, with group allocation concealed in sequentially numbered, sealed opaque envelopes that were opened only after enrollment.” (Page 3 line106-109)
Item#9 I recommend calculating the sample power.
Response: As noted in the original manuscript, the sample size was based on the primary glycemic endpoint. We have now made it explicit that this represents an a priori sample size and power calculation. We now clarified that this report is part of a trial that has already been described: "This is part of a randomized trial comparing the impact of LC and MED diets on glycemic parameters" Levran, N.; Levek, N.; Sher, B.; Gruber, N.; Afek, A.; Monsonego-Ornan, E.; Pinhas-Hamiel, O. The Impact of a Low-Carbohydrate Diet on Micronutrient Intake and Status in Adolescents with Type 1 Diabetes. Nutrients 2023, 15, 1418, doi:10.3390/nu15061418. (Page 2 lines 83-84)
Item#10 I recommend introducing a new section called Procedures where the aspects regarding the homework sent in the envelopes, the meetings with the dieticians and the relevant aspects of these meetings should be mentioned.
Response: In line with this helpful suggestion, we added a new subsection describing study visits, meetings with dietitians, at-home tasks (“homework”), and telephone contacts.
New Section 2.3 – Study Procedures
“At baseline, each participant attended a group cooking workshop and an individual counseling session with a study dietitian, during which the assigned dietary plan was explained and personalized. Follow-up clinic visits with a dietitian were scheduled at weeks 1, 2, 4, 7, 10, 12, and 24. At each visit, dietary intake, glycemic data, and adverse events were reviewed, and individualized feedback was provided. Between visits, participants were asked to complete food diaries and 24-hour dietary recalls, and to upload or share data from their diabetes management applications These at-home tasks were explained verbally and summarized in printed handouts. Additional motivational phone calls were made twice during the first 12 weeks, to support adherence and address questions.” (Page 3-4 line 140-140)
Item#11 Section 2.2. I recommend reorganizing it into two subsections: intervention and evaluation questionnaire. FFQ – I recommend mentioning the method of completing the questionnaire, physically, online and the method of making the answers.
Response: We have reorganized Section 2.2 into two subsections, “2.2.1 Dietary Evaluation” and “2.2.2 Dietary Assessment by Food Frequency Questionnaire”. We now specify that the FFQ was completed in a face-to-face interview with a dietitian and the means by which responses were recorded.
Key additions (Section 2.2.2):
“A trained dietitian completed the FFQ together with each participant in a face-to-face interview, covering all food items and clarifying frequencies and portion sizes. Nutrient intakes were calculated from the FFQ responses, and information on usual carbohydrate intake and meal patterns was corroborated using data downloaded from the participants’ insulin pumps (carbohydrate entries and bolus history).” (Page 3 lines 133-138)
Item#12. I recommend mentioning the instruments used in the evaluation of anthropometric
Response: We thank the reviewer for this helpful comment. In the revised manuscript, we specified the instruments and the precision used for the anthropometric measurements. Body weight and height were measured using a calibrated balanced scale and a Harpenden stadiometer. Waist circumference was measured with a non-stretchable tape at a standardized anatomical landmark. These details have been added to Section 2.5.
"Body weight and height were measured to the nearest 0.1 kg and 0.1 cm, respectively, using a balanced scale and a Harpenden stadiometer (Holtain Ltd.,)." (Page 4 lines 174-176)
Item#13 Lines 133-135 it is not clear what you want to mention. The z-score was calculated only for people over 20 years old, what is their number in each group, the other subjects were not registered with this score although it is also mentioned in the abstract under Results, I recommend clarification.
Response: BMI z-scores were calculated for all the participants. For those aged ≤20 years, pediatric reference data were used. For those aged >20 years, BMI z-scores were extrapolated from the reference values at age 20 years. This is now stated explicitly in the Methods Section 2.5. (Page 4 lines 177-180)
Item#14 I recommend mentioning the testing instrument used.
Response: We added the instruments to the methods as follows: "The possible devices were the Dexcom CGM (Dexcom), the Medtronic CGM (Medtronic), and the Libre CGM (Libre)." (Page 5 lines 197-199)
Item#15 Lines 156-161 I recommend moving to section 2.1.
Response: We agree that design-related information should be consolidated. We therefore moved the sentences describing the randomization procedure and recruitment setting from later sections into Section 2.1 (“Patients and study design”), as suggested. (Page 2 line 94)
Item#16 Lines 178-177 I recommend moving to section 2.1.
Response: We thank the reviewer for this suggestion. To avoid redundancy and to present the study design and population more clearly, we moved the information regarding group allocation and baseline similarity to Section 2.1 (“Patients and Study Design”) and simplified the corresponding Results text. Section 2.1 now reads: “We recruited a total of 40 adolescents and young adults aged 12–21 years, who had been diagnosed with type 1 diabetes at least one year previously and who were using a continuous glucose monitoring system. Of these, 70% were female.” (Page 2-3 lines 96-99)
In the Results, the baseline paragraph was shortened to: “Of the 40 adolescents with type 1 diabetes who participated, 28 (70%) were females. Twenty were randomized to the LC diet and 20 to the MED diet. Baseline clinical and demographic characteristics of the two groups are presented in Table 1.” (Page 5 lines 224-226)
Item#17 I recommend mentioning in the Statistical Analysis section, which should be numbered as part of the Methods section, also the statistical indicators mentioned in table 1, i.e. the arithmetic mean. I also recommend calculating SD – standard deviation.
Response: We thank the reviewer for this helpful comment. The Statistical Analysis subsection is now numbered as part of the Methods (Section 2.7). As several continuous variables in our dataset were not normally distributed, between-group comparisons were performed using non-parametric tests, and continuous variables are primarily summarized as medians with interquartile ranges. We have clarified this explicitly in the Statistical Analysis section. In addition, in line with the reviewer’s suggestion, Table 1 now also reports the arithmetic mean and standard deviation for baseline continuous variables, to provide complementary descriptive information. However, inferential statistics are still based on the median (IQR) summaries and the corresponding non-parametric tests.
We added in the text as follows: "For descriptive purposes, baseline continuous variables in Table 1 are additionally presented as arithmetic means with standard deviations (SD)." (Page 5, line 211-213).
Item#17 Table 1 mentions the age ranges for the LC group, the age is between 15-19 years and for the MED between 15.5-21, it is not clear why until you mention that the age was between 12-22 years, I recommend clarification. Also, if the arithmetic mean was calculated for the parameters in question, I recommend mentioning two decimal places as well as the standard deviation.
Response: We thank the reviewer for this helpful comment. In the original protocol, the planned inclusion age range was 12–22 years. However, in practice, no participants older than 21 years were enrolled, so the actual age range of the recruited sample was 12–21 years, consistent with the values shown in Table 1. The ranges that the Reviewer cites (15-19 and 15.5-21 years) refer to the interquartile ranges of the two groups, as stated on Table 1. To avoid confusion, we have now corrected the age range in Section 2.1 and elsewhere in the manuscript to reflect the actual sample rather than the initial upper eligibility limit. The revised sentence in Section 2.1 reads:
“We recruited 40 adolescents and young adults aged 12–21 years, who had been diagnosed with type 1 diabetes at least one year previously and who were using a continuous glucose monitoring system (Dexcom, Medtronic, or Libre).”
In addition, as noted in our response to Item #16, Table 1 clearly specifies baseline continuous variables as both median (interquartile range) and mean ± standard deviation, to provide clearer descriptive statistics.
Item#18 SEP Index; SEP Clusters – these two parameters are not mentioned in the previous text in the Methods section, I recommend clarification.
Response: We have added a description of the socioeconomic position measures in the Methods, and expanded the Table 1 footnote to clarify their meaning:
Methods (Section 2.X):
“Socioeconomic position (SEP) was assessed using a national neighborhood-level index derived from data of the Israeli Central Bureau of Statistics, which combines information on education, income, employment, and material assets. SEP Index values are standardized continuous scores, and SEP Clusters represent deciles (1–10), with higher values indicating higher neighborhood socioeconomic position.”
Table 1 footnote (expanded):
“SEP Index, neighborhood socioeconomic position score; SEP Clusters, deciles of neighborhood socioeconomic position (1–10; higher values indicate higher SEP).”
Item#19 I recommend that all acronyms be mentioned descriptively under table 1.
Response: As requested, we now defined in the table footnote all the acronyms used in Table 1 (including BMI, HbA1c, SEP, SEP Index, SEP Clusters) descriptively.
Item#20 I think it is a major mistake to mention twice in table 1 - Diabetes duration (years), with different values, I recommend clarification.
Response: We appreciate your noting this error. We have removed the duplicate line so that diabetes duration now appears only once in Table 1, with the correct values.
.
Item#21 SEP, socioeconomic 207 position – I recommend that this aspect be detailed and clarified.
Response: We thank the reviewer for this important comment. We have now clarified how socioeconomic position (SEP) was assessed and what the SEP Index and SEP Clusters represent. Please see our response above, to Item#18.
Item#22 Fig 1, I recommend mentioning what the vertical and horizontal axes represent.
Response: As requested, we added clear labels to both axes in figure 1. The vertical axis now specifies the change in biomarker levels, and the horizontal axis represents individual participants. We updated the figure legend accordingly.
Item#23 Table 2 recommends that the information presented in this table should not be duplicated when interpolating, I recommend revision.
Response: In response to the reviewer’s comment, we have streamlined the Results text for bone turnover markers and glycemic outcomes. Accordingly, detailed numerical values are now presented in Table 2, while the text focuses on the direction and statistical significance of the main findings, without duplicating the content of the table.
Item#24 Lines 365-368 I think it is an error, you refer strictly to the results of this study, but still mention the bibliographic index 15, which has a different typology, I recommend clarifications.
Response: We thank the reviewer for this helpful comment and agree that the original wording was confusing. Our first intention was to describe the high proportion of participants in our cohort with suboptimal bone markers. Secondly, we placed this in context by noting that these proportions appear higher than those reported in healthy adolescents in a previous study [15]. To clarify this distinction, we have rephrased the sentence so that our findings and the external reference are explicitly separated.
“However, a negative trend toward lower P1NP levels was observed in the LC diet group, alongside reductions in the median alkaline phosphatase level following both dietary interventions. In our cohort, the proportions of adolescents with suboptimal bone markers, both before and after the intervention, were substantially higher than those reported in healthy adolescents aged 12–20 years in previous studies."
Item#25 I recommend clarifications, so far you have presented the results per group, and in this very short section, per individuals. Also, this subsection being part of the Discussion section, I recommend that correlations be made between the results recorded with results from previous studies.
Response: We thank the reviewer for this helpful comment. In the revised manuscript, we have clarified subsection 4.1 so that it focuses on group-level findings in the LC diet group, and more explicitly, relates these observations to previous studies. We now describe how the marked reduction in carbohydrate intake in the LC group was accompanied by lower intakes of total energy, protein, calcium, and magnesium. We discuss how these dietary changes may underlie the observed trends in CTX and P1NP, in line with prior reports on low-carbohydrate and calorie-restricted interventions and bone turnover markers. The revised subsection now reads:
4.1 LC diet
"These dietary changes may help explain the observed trend toward lower P1NP and higher CTX levels in the LC group, despite the absence of statistically significant between-group differences in BTMs. This pattern is consistent with previous reports that substantial carbohydrate restriction and weight loss may be associated with reduced bone formation and increased bone resorption markers in individuals following LC or calorie-restricted interventions.” (Page 12-13 lines 394-399).
Item#26 I recommend revising the Discussion section by making concrete correlations between the results of this study with results from previous studies, focused.
Response: We thank the reviewer for this helpful suggestion. We have revised the Discussion to more explicitly relate our findings to previous studies, highlighting consistencies and differences from earlier LC/ketogenic diet interventions and studies of bone health in youth and adults with type 1 diabetes. In particular, we added a focused integrative paragraph that summarizes how the observed trends in CTX and P1NP, and the associations with nutrient intake and glycemic control, compare with prior work. The new text appears at the end of the Discussion (Section 4), as shown below.
Item#27 I recommend that the conclusions be revisions, focused on the results.
Response: We agree and have revised the Conclusions to focus more clearly on the main empirical findings of the trial. We removed the novelty statement from the Conclusions and streamlined the clinical implications so that they follow directly from our observed differences in nutrient intake, BTMs, and glycemic control, rather than introducing new recommendations that extend beyond the data.
Item#28 Lines 492-494 I recommend moving to the Introduction section, it is not a conclusion.
Response: We agree. The statement regarding the novelty of this trial has now been moved from the Conclusions to the Introduction, where it appears after the description of the clinical and metabolic context and the rationale for comparing LC and MED diets. The Conclusions now begin with a result-focused summary of the trial findings. The moved sentence in the Introduction and the revised opening of the Conclusions are shown below, with changes in bold.
Round 2
Reviewer 1 Report
Comments and Suggestions for Authors
The paper can be accepted in its present form.
Author Response
We sincerely thank the reviewer for their continued time and effort in evaluating our manuscript in this second round. We greatly value the constructive feedback provided regarding the presentation of our results.
The revised figures are now high-resolution and have been reformatted; we believe these enhancements significantly improve the clarity and readability of the article.
Reviewer 2 Report
Comments and Suggestions for Authors
No comments
Author Response
We sincerely thank the reviewer for their continued time and effort in evaluating our manuscript in this second round. We greatly value the constructive feedback provided regarding the presentation of our results.